# Automatic mapping of high-risk urban areas for *Aedes aegypti* infestation based on building facade image analysis

**Camila Laranjeira**[1☉], **Matheus Pereira**[1☉], **Raul Oliveira**[2], **Gerson Barbosa**[3], **Camila Fernandes**[2], **Patricia Bermudi**[2], **Ester Resende**[1], **Eduardo Fernandes**[1], **Keiller Nogueira**[4], **Valmir Andrade**[5], **José Alberto Quintanilha**[6], **Jefersson A. dos Santos**[1,7‡], **Francisco Chiaravalloti-Neto**[2‡]*

**1** Department of Computer Science, Universidade Federal de Minas Gerais, Belo Horizonte, Brazil, **2** Department of Epidemiology, School of Public Health of University of São Paulo, São Paulo, Brazil, **3** Pasteur Institute, Secretary of Health of the State of São Paulo, São Paulo, Brazil, **4** Computer Science and Mathematics, University of Stirling, Stirling, United Kingdom, **5** Epidemiologic Surveillance Center, Secretary of Health of the State of São Paulo, São Paulo, Brazil, **6** Institute of Energy and Environment, University of São Paulo, São Paulo, Brazil, **7** Department of Computer Science, University of Sheffield, Sheffield, United Kingdom

☉ These authors contributed equally to this work.
‡ These authors also contributed equally to this work.
* franciscochiara@usp.br

**Data Availability Statement:** The source code and trained models are available on GitHub at https://

## Abstract

### Background

Dengue, Zika, and chikungunya, whose viruses are transmitted mainly by *Aedes aegypti*, significantly impact human health worldwide. Despite the recent development of promising vaccines against the dengue virus, controlling these arbovirus diseases still depends on mosquito surveillance and control. Nonetheless, several studies have shown that these measures are not sufficiently effective or ineffective. Identifying higher-risk areas in a municipality and directing control efforts towards them could improve it. One tool for this is the premise condition index (PCI); however, its measure requires visiting all buildings. We propose a novel approach capable of predicting the PCI based on facade street-level images, which we call PCINet.

### Methodology

Our study was conducted in Campinas, a one million-inhabitant city in São Paulo, Brazil. We surveyed 200 blocks, visited their buildings, and measured the three traditional PCI components (building and backyard conditions and shading), the facade conditions (taking pictures of them), and other characteristics. We trained a deep neural network with the pictures taken, creating a computational model that can predict buildings' conditions based on the view of their facades. We evaluated PCINet in a scenario emulating a real large-scale situation, where the model could be deployed to automatically monitor four regions of Campinas to identify risk areas.

github.com/patreo-lab/PCINet. Sampled data cannot be made openly available due to it containing sensitive information regarding the condition of buildings from the inhabitants of Campinas. Readers that want to collect the protected data should email the Ethics Committee of School of Public Health of Univerity of São Paulo (email: coep@fsp.usp.br).

**Funding:** This work was supported by the São Paulo Research Foundation - FAPESP (grant 2020/01596-8 to FCN); by the National Council for Scientific and Technological Development - CNPq (grant 304391/2022-0 to FCN); by the Serrapilheira Institute (grant R-2011-37776 to JAS); by the Minas Gerais State Research Support Foundation – FAPEMIG (grant APQ-00449-17 to JAS) and CNPq (grant 306955/2021-0 to JAS and grant 305188/2020-8 to JAQ). This work was also supported by FAPESP (grant 2020/12371-7 to PMMB). The funders had no role in study design, data collection and analysis, decision to publish, or preparation of the manuscript.

**Competing interests:** The authors have declared that no competing interests exist.

## Principal findings

PCINet produced reasonable results in differentiating the facade condition into three levels, and it is a scalable strategy to triage large areas. The entire process can be automated through data collection from facade data sources and inferences through PCINet. The facade conditions correlated highly with the building and backyard conditions and reasonably well with shading and backyard conditions. The use of street-level images and PCINet could help to optimize *Ae. aegypti* surveillance and control, reducing the number of in-person visits necessary to identify buildings, blocks, and neighborhoods at higher risk from mosquito and arbovirus diseases.

## Author summary

The strategies to control *Ae. aegypti* require intensive work and considerable financial resources, are time-consuming, and are commonly affected by operational problems requiring urgent improvement. The PCI is a good tool for identifying higher-risk areas; however, its measure requires a high amount of human and material resources, and the aforementioned issues remain. In this paper, we propose a novel approach capable of predicting the PCI of buildings based on street-level images. This first work combines deep learning-based methods with street-level data to predict facade conditions. Considering the good results obtained with PCINet and the good correlations of facade conditions with PCI components, we could use this methodology to classify building conditions without visiting them physically. With this, we intend to overcome the high cost of identifying high-risk areas. Although we have a long road ahead, our results show that PCINet could help to optimize *Ae. aegypti* and arbovirus surveillance and control, reducing the number of in-person visits necessary to identify buildings or areas at risk.

## 1 Introduction

### 1.1 Mosquitoes, arboviruses, and higher-risk areas

A myriad of known viruses have arthropods as vectors, of which 30 are known to cause disease in humans [1]. Even with this diversity, four viruses significantly impact human health, causing yellow fever, dengue, Zika, and chikungunya. The commonality among these diseases is that female *Aedes* mosquitoes transmit their viruses. Historically, the most important is *Aedes aegypti*, which is linked to the spread of dengue epidemics [2] and responsible for yellow fever epidemics in the past. *Ae. aegypti* is also involved in the explosive epidemics of chikungunya (alphavirus) [3] and Zika (flavivirus) [4], which reinforces its role as a vector of diseases with increasing importance in the Americas and the entire world. During 2019, a dengue outbreak spread widely throughout the Americas, causing more than 2.3 million infections in Brazil alone [5].

   Of these four arboviruses, we have an effective vaccine against the yellow fever virus. A promising vaccine against the dengue virus, named Qdenga, has recently emerged, which was approved for a broader audience and does not require prior exposure. It is worth noting that this vaccine is initially available only through private laboratories [6]. Considering this scenario, the prevention of infections transmitted by *Ae. aegypti* will continue to rely on

decreasing contact with it and developing control measures against its immature (larvae and pupae) and adult forms, mainly the females, which feed almost exclusively on human blood [7, 8].

*Ae. aegypti* is quite prevalent in urban areas, where it uses artificial and natural containers with water to reproduce. In urban environments, the large presence of containers capable of accumulating water creates environments conducive to the reproduction of mosquitoes, which is one of the reasons for the failure of many attempts to control the diseases and their vector, *Ae. aegypti* [9, 10]. The strategies to control *Ae. aegypti* require intensive work and large financial resources, are time-consuming, and are commonly affected by operational problems. Moreover, several studies have shown that strategies currently used in control programs are not sufficiently effective, or even ineffective, and require urgent improvement [11–19].

Different approaches have been used to guide policies to fight dengue, Zika, and chikungunya in large cities. In endemic areas, notably Latin America, Southeast Asia, and the Pacific, most surveillance relies on traditional methods, such as health service reports and laboratory confirmation of a subset of cases to a central health agency [20]. Although this approach has some accuracy, its effectiveness is hindered by the significant time gap between case detection and notification to the system [21]. This delay restricts the responsiveness of health authorities, impeding the implementation of prompt and effective responses and resulting in severe consequences [20]. In addition to this problem of case detection and notification, preventive surveillance and control for *Ae. aegypti*, a crucial strategy at the level of public policies, also faces difficulties that reduce its effectiveness. *Ae. aegypti* entomological surveillance and control involve great effort for health services and high costs for developing house-to-house vector monitoring [22–24]. It is also time-consuming to manually aggregate and validate all data [25].

Because different places could have different *Ae. aegypti* infestation levels, one way to improve arbovirus surveillance and control is to identify the buildings, blocks, or neighborhoods with higher risks in a municipality. One of the tools that can be used to direct *Ae. aegypti* surveillance and prevention efforts to higher-risk areas is the Premise Condition Index (PCI) [26]. This was proposed by Tun-Lin *et al.* [26, 27] and considers in its scope the place conditions, conservation, and shading, assigning a score on a scale that indicates a greater propensity of a given building to become a breeding ground for *Ae. aegypti* mosquitoes. Several studies have tested this relationship and have found similar results. In a survey conducted in the city of Rio de Janeiro, it was observed that the number of *Ae. aegypti* eggs was higher as the PCI increased [28]. This same association was observed in Botucatu, in the state of São Paulo [29] and in Campos dos Goytacazes, in the state of Rio de Janeiro [28], Brazil. In Marília in the state of São Paulo, a positive relationship was observed between PCI and the presence of larvae and pulps in the buildings evaluated [30]. A study conducted in Campinas in the state of São Paulo, which also confirmed this relationship, proposed the adoption of an extended PCI considering other variables such as backyard paving, the existence of *Ae. aegypti* potential breeding sites, and the presence of animals in the buildings [31].

The issues with applying PCI to identify risk areas are the same as those of other strategies, that is, intensive work and high costs, and it is time-consuming. In this study, we hypothesized that using facade street-level images and artificial intelligence (AI), we could predict the PCI of buildings without developing house-to-house monitoring. Adopting computational methods and utilizing AI could address the presented challenges, offer a substantial and cost-effective advancement to inform public policies, and enhance the effectiveness of *Ae. aegypti*-related disease monitoring and prevention [32, 33].

### 1.2 Artificial intelligence applied to the problem

The preventive initiatives currently carried out in the fight against arboviruses mainly focus on mapping and preventing the spread of disease vectors. There are many ways one can leverage AI, specifically machine learning, in this scenario, both from the perspective of which data to gather (i.e., the input) and which measures to estimate (i.e., the output).

From a data perspective and considering that some studies have shown that vulnerable urban areas have higher *Ae. aegypti* infestation levels [34–37], it is common to use field survey data related to socioeconomic status [38, 39], such as income, education, and crowding. There are also instances of leveraging environmental information, such as temperature [40, 41], humidity, or precipitation [42]. We are especially interested, however, in the domain of images, which has received growing attention from the research community. At the same time, Lorenz *et al.* [43] showed that information extracted from aerial images can be positively correlated with mosquito infestation, with many studies following the same data path [44–46]; however little attention has been given to the abundance of information one can extract from facade images, which is the main focus of the present work.

Regarding target inferences, directly predicting mosquito infestation has a significant drawback: data collection is highly cumbersome as it depends on house-to-house visits and/or physically installing and monitoring traps. Machine learning approaches can benefit from large volumes of data; thus, it is possible to find works resorting to proxy tasks that allow faster and/or cheaper data gathering, which can better scale to broader geographical regions. The review presented by Joshi and Miller [47] shows that one of the most prominent proxy tasks is locating common mosquito breeding grounds, such as tires, buckets, and water tanks, to name a few, reframing the problem as an object detection task. Works such as that of Cunha *et al.* [45], who detected swimming pools and water tanks, also mention the correlation of such breeding sites with socioeconomic status.

Looking at the problem from a novel perspective, the work of Zou *et al.* [48] is worth mentioning. Although it was not directly applied to disease control, the authors showed that signs of building abandonment can be better derived from facade images since an aerial view will always be limited to the building's roof and surrounding area. Building abandonment, or lack of maintenance, has many similarities to PCI, as they are both interested in visual cues, such as overgrown vegetation, and wall deterioration. We could not find any work directly inferring PCI from facade images in the literature. Thus, we provide novel contributions to the literature by applying a state-of-the-art deep learning-based method to this task.

### 1.3 Objectives

In this work, we propose a novel approach capable of predicting the PCI of buildings based on street-level images. This is the first work combining deep learning-based methods with street-level data to predict PCI, an essential indicator of *Ae. aegypti* infestation. This study is part of the project granted by the São Paulo Research Foundation (FAPESP—process 2020/01596-8), entitled "Use of remote sensing and artificial intelligence to predict high-risk areas for *Aedes aegypti* infestation and arbovirus", named here as our entire project.

## 2 Related work

Due to its social and health-related relevance, several different techniques [43–46, 49–51] have been proposed to combine AI with image processing toward the mapping of *Ae. aegypti* risk areas. Albrieu *et al.* [44] classified 32 neighborhoods into 17 environmental classes extracted from SPOT 5 satellite data. Then, they correlated such classification with data from entomological surveys and analyzed which characteristics are most related to the proliferation of *Ae.*

*aegypti*. Kim *et al*. [51] combined Normalized Difference Water Index with the rectangular fit space metric [52] to detect *Culex* mosquito breeding sites (such as swimming pools) in satellite imagery and consequently helped to control the population of this West Nile virus vector. Andersson *et al*. [49] proposed new deep learning-based networks capable of predicting dengue fever and dengue hemorrhagic fever rates in a certain area based on street-level imagery surrounding that region. Lorenz *et al*. [43] exploited machine learning techniques to perform pixel-wise land-cover classification using satellite images of one specific study area. After classifying pixels into ten possible classes (such as asphalt, asbestos roof, exposed soil, and water), they conducted an analysis correlating this information with mosquito data collected using traps to identify the physical characteristics of a landscape that most influence the distribution of *Ae. aegypti* adult mosquitoes.

More recently, Andersson *et al*. [50] proposed a new network that fuses information extracted from aerial data and street-level images to identify environmental factors linked to *Ae. aegypti* mosquitoes and, consequently, predict dengue fever rate in urban scenarios. Haddawy *et al*. [53] explored a detection network to identify dengue vector breeding sites (such as buckets, old tires, and potted plants) in street view images. To allow better observation and understanding of the region, they used several images to cover the entire surroundings of the area. Lee *et al*. [54] combined entomological and health-related data with information extracted from Unmanned Aerial Vehicle images (such as water containers, and green-red vegetation index) to identify high-risk rural areas of mosquito infestation. Liu *et al*. [55] compared and combined environmental features extracted from street-level images using pretrained networks with standard features (such as epidemical, meteorological, and sociodemographic variables) to create a machine learning model capable of performing weekly dengue forecasting. They concluded that incorporating environmental data from street view images makes the model more effective for predicting urban dengue. Cunha *et al*. [45] employed a deep learning model to detect water tanks and swimming pools in aerial data. Based on this detection, they conducted an analysis correlating the number of water tanks and swimming pools with the socioeconomic levels of the different regions, finding that areas with low socioeconomic status had more exposed water tanks, while regions with high socioeconomic levels had more exposed pools. They argued that these results could help to identify *Ae. aegypti* higher-risk areas as there is a positive relationship between infestation and vulnerable areas [34–37]. Passos *et al*. [46] combined convolutional network-based models with the spatiotemporal tube concept [56] to integrate spatial and temporal data, thus allowing the detection of water tanks and tires (the most reproductive containers for *Ae. aegypti* species [47]) in aerial videos.

## 3 Materials and methods

### 3.1 Ethics statement

The present study was approved by the Research Ethics Committee of the School of Public Health at the University of São Paulo, in the Plataforma Brasil system, Ministry of Health, number CAAE: 46655121.0.0000.5421; May 21, 2021.

### 3.2 Description of the study area

The city of Campinas (22°53'03" S and 47°02'39" W) has the third largest population in the state of São Paulo, with just over one million inhabitants living in an area of 794, 571$km^2$, with a good index of human development (0.805). Its area was divided by the Brazilian Institute of Geography and Statistics (IBGE) into 1695 urban and 54 rural census tracts for conducting the 2010 demographic census (Fig 1). Campinas is located in a metropolitan region with

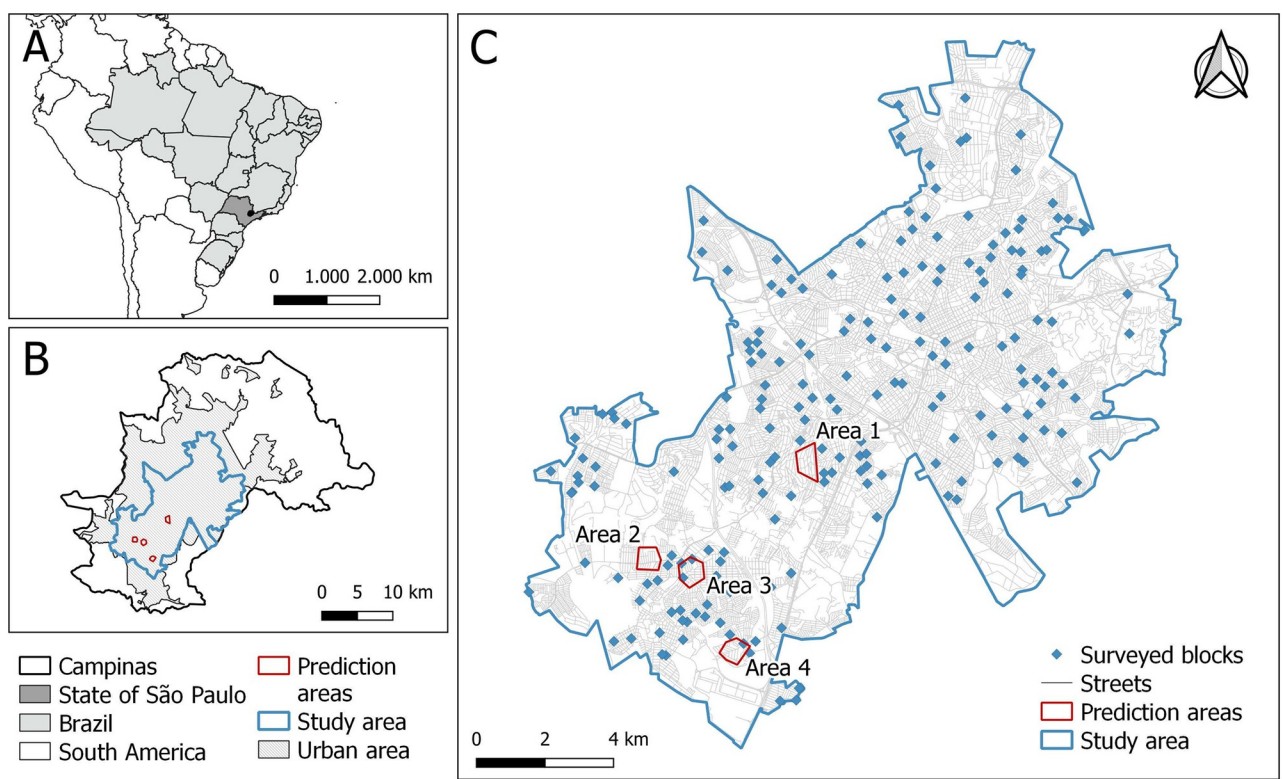

**Fig 1.** (A) Map of Campinas, state of São Paulo, Brazil, South America; (B) Campinas municipality: total, urban and study areas; (C) Study area, with the localization of 200 surveyed blocks and the four areas used to predict the PCI facade. Base layers of the map: https://www.ibge.gov.br/geociencias/downloads-geociencias.html?caminho=recortes_para_fins_estatisticos/malha_de_setores_censitarios/censo_2010/base_de_faces_de_logradouros_versao_2021 and https://www.ibge.gov.br/geociencias/organizacao-do-territorio/malhas-territoriais/15774-malhas.html?=&t=acesso-ao-produto.

approximately 3.3 million inhabitants. It has a hot and temperate climate, characterized by an average annual temperature of 19.3˚C and an average annual rainfall of 1, 315*mm*. The city has been infested with *Ae. aegypti* since 1991, and dengue transmission has been observed in the municipality since 1996. Since then, there has been an expansion of transmission areas and an increase in reported cases, with approximately 175 thousand dengue cases reported from 2010 to 2023. The Ministry of Health classifies the municipality as a priority due to its incidence of infection and geographic location. It is connected by several roads with an intense flow of vehicles, has an international airport, and an intense flow and movement of people, increasing the possibility of arbovirus transmission and spreading to other areas of the state and country. These factors, together with its vast territorial expanse and heterogeneity in infrastructure, land use, and lifestyle habits, contribute to the municipality's vulnerability to arboviruses. Campinas experienced two major dengue epidemics in 2014 and 2015, with 42,109 and 65,209 cases recorded, respectively. The first autochthonous Zika cases were reported in the city in 2016. The Department of Health Surveillance of the Municipality Health Department of Campinas reported 11,268 cases of dengue in 2022, equivalent to an incidence rate of 923.6 per 100,000 residents, with the highest cases occurring in March and May, and 19 confirmed cases of Chikungunya. For our entire project, we considered that the study area was composed of 1293 Campinas urban census tracts (Fig 1) covered totally or partially by the high-resolution satellite image granted by FAPESP.

### 3.3 Data collection and database structuring

**3.3.1 Sampling of sectors and blocks.** We used the following criteria to consider an urban census tract eligible for conducting the field measurement of PCI: a census tract with more than 90% of their area contained within the study area; with the São Paulo Social Vulnerability Index (IPVS) classification, developed by the State Data Analysis System Foundation (SEADE); and with 20 or more households. With these criteria, we obtained 1054 census tracts, and the sampling was conducted through a systematic random draw. For this, initially, a database containing the codes list of the study area census tracts was created, ordered by the socioeconomic and demographic factor values of IPVS; the proportion of houses among buildings; and average temperatures. IPVS measures social inequality within municipalities and serves as a parameter for the development of specific public policies. These variables were chosen due to the known positive relationship between *Ae. aegypti* and arbovirus diseases, and higher average temperature [34–37, 57–60]. Temperature data were collected from the Moderate-Resolution Imaging Spectroradiometer (MODIS) satellite image dataset [61]. The factors of the IPVS were obtained from SEADE. The proportion of houses among buildings and the list of sector codes were taken from IBGE [62]. Then, we systematically selected a sample with 200 census tracts, using a ratio of 5.27 (1054/200).

Of these 200 chosen final census tracts, two groups of 100 were allocated, alternately, for the first and second moments of fieldwork, as detailed below. Regarding the representative blocks, for each census tract, one that was considered adequate was selected. For this choice, we aimed to measure the PCI at least in 10 buildings, taking into account that, in Campinas, according to vector control agents, there is a refusal rate of approximately 40 to 50% visits.

**3.3.2 Field data collection and database.** To evaluate the PCI, the model by Tun-Lin *et al.* [26] and the extended model by Barbosa *et al.* [31] were used, adding other variables to improve the classification of the building and expanding the score (1 to 5, instead of 1 to 3). We used the following characteristics: building type, facade, building and backyard conditions, shading, backyard paving, roofing, and potential breeding sites. Contrary to these studies, where level 1 indicated the best and 3 the worst condition, we considered level 1 to indicate the worst condition and 5 the best, seeking less subjectivity in the classification, as follows:

**Building Type**: 1-House; 2-Commerce; 3-Industry; 4-Apartment building; 5-Others (church, school, etc.).

**Facade or Building condition**: 1-Facade or building built in wood or a material other than masonry, lack of internal paving, and restricted access to basic sanitation; 2-Facade or building built in masonry and without plaster, or finished facade or building with at least five signs of lack of maintenance, with little or no access to basic sanitation; 3-Facade or building built in masonry with only plaster, with access to basic sanitation, or finished facade or building with two signs of lack of maintenance; 4-Finished facade or building, but with some sign of lack of maintenance; 5-Finished facade or building with no signs of lack of maintenance.

**Signs of lack of maintenance**: Old, peeling paint; Mold and mildew spots on the walls; Vegetation with disordered growth; Dry vegetation; Rust on gates and/or window; Broken windows; Old mail in mailboxes or gates; Cracked and/or broken walls; Presence of useless items, garbage, or advertisements; Rusty padlocks and chains; Graffiti; Broken or cracked pavement.

**Backyard Condition**: 1-Very poorly maintained (with garbage, fallen leaves, animal waste—disorganized); 2-With little care (with garbage, fallen leaves and/or animal waste—poorly organized, in an intermediate situation between 1 and 3); 3-With average care (little garbage, fallen leaves and/or animal waste—poorly organized); 4-Reasonably well maintained (very little litter, fallen leaves and/or animal waste—reasonably organized and in an intermediate

situation between 3 and 5); 5-Very well maintained (no garbage, no fallen leaves and no animal waste, organized).

**Shading**: 1-Fully shaded backyard (shade from trees and plants, neighboring buildings, walls, etc.); 2- Backyard 2/3 shaded (shade from trees and plants, neighboring buildings, walls, etc.); 3-Backyard 1/3 shaded (shade from trees and plants, neighboring buildings, walls, etc.); 4- Backyard without shading (shade from trees and plants, neighboring buildings, walls, etc.); 5-Land fully built.

**Backyard paving**: 1-Backyard without paving; 2-Backyard 25% paved; 3-Backyard 50% paved; 4-Backyard 75% paved; 5-Fully built land (no backyard) or fully paved backyard.

**Roofing**: 1-Without tiles or other coverage (canvas, plastic, plywood, etc.); 2-Asbestos/Zinc tile or slab; 3-Clay tile, cement or building covering.

**Potential breeding cites**: Presence or not of containers that are potential breeding grounds for *Ae. aegypti*.

Fieldwork was conducted in two stages: 100 blocks from September to November 2021, and 100 blocks from March to May 2022. Between these two periods, we developed other fieldwork to achieve all project objectives, such as mosquito collections with adult traps.

For data collection in the field with the aim of measuring PCI, an app was developed for the Android operating system and installed on a 9-inch tablet. This system was conceived to facilitate the digital collection of data and automatically obtain the coordinates of each building visited and allow taking pictures of the facades of the buildings. Before the start of activities, the field team was trained to classify the buildings into the PCI characteristics and to use the equipment. The data collected in the field were stored offline on the tablet and later downloaded via a Wi-Fi connection to a PostgreSQL database, not requiring a data package. The data were later exported in CSV format, along with the images for analysis.

**3.3.3 Database merging and treatment.** By merging the data acquired from the two field collection procedures, we produced a CSV file containing 5329 lines and a total of 7785 images. However, this data contained errors due to collection problems (such as corrupted image files) or errors later acquired during the initial data processing. For this reason, we performed additional filtering on this data with the objective of leaving the final dataset with less noise or undesirable conditions. This process was conducted in five steps, as follows:

1. First, we removed duplicate images. From the set of 7785 initial images, we noticed that some were duplicates, i.e., they presented the same pixels but in different files. We ran a script to compare all pairs of images, leaving just one from each set of repeated images. After this procedure, 3469 images were discarded, remaining 4316.

2. The second step involved deleting corresponding lines from the CSV that pointed to images discarded in step 1, as each visit should be paired with a unique image. After this procedure, the 5329 initial lines were reduced to 4190, as some removed images had no corresponding line in the CSV.

3. We also noticed that some lines from the CSV pointed to the same geographical coordinates. To avoid duplicates or buildings with varying PCI scores in the same dataset, we also removed all but one (for each case) of the lines that pointed to repeating coordinates. This reduced the number of lines in the CSV from 4190 to 4172.

4. With the remaining images from step 1, we performed a manual verification to remove the ones that were not adequate for training due to problems during the capturing of the photo. These are images in which the property's facade is not visible, such as photos pointing to wrong directions or with objects blocking the view. In this step, 108 images were discarded.

5.  Finally, we matched the remaining lines of the CSV with their corresponding images from the set that remained after the replacedfourthfirst step. Thus, we discarded the images that did not have a corresponding line in the CSV pointing to them. This left the final dataset with 4060 valid pairs of images and lines in the CSV.

**3.3.4 PCI dataset descriptive analysis.**   For an overview of the collected dataset, Section 4.1 presents the characteristics of surveyed buildings in terms of each PCI attribute with its correspondent distribution and correlation with the target label. Distributions are presented as relative percentages, while correlation is calculate with the Spearman correlation metric.

**3.3.5 Street view data collection.**   Although photographs from the building facades can be taken through fieldwork, this process requires human work and takes time, making it difficult to escalate to more extensive regions. Ideally, an automatic way of quickly gathering images for the buildings' facades should be employed, allowing for the processing of many neighborhoods or even cities in a short time. Google Street View is a good alternative in the presented context, as its API allows for the collection of images from urban environments around different parts of the world, making it possible to aim their views towards building facades.

There is a natural difference between images acquired from human fieldwork and large-scale sources, such as the aforementioned Google Street View, because the type of sensor or camera used can change the characteristics of the data. This can be enough to make a computational model trained on one type of imagery unable to work with the other type correctly. Thus, to validate our models and test their capabilities to work with high-scale sources, we collected data from Google Street View.

We collected Street View images with the use of Google's Street View Static API. For this, we required the coordinates of each building of interest. In this work, the coordinates were manually (in person) taken from four regions of Campinas with varying socioeconomic characteristics (Fig 1). Area 1 (with a higher socioeconomic level), according to the 2010 census of IBGE, had an average income of 1807.00 Reais (the Brazilian currency) and 3.0 inhabitants per household; areas 2 and 3 (with intermediate socioeconomic level) had average incomes of 1285.00 and 1138.00 Reais, with 3.2 and 3.4 inhabitants per household, respectively; and area 4 (with lower socioeconomic level) had an average income of 755.00 Reais and 3.5 inhabitants per household [45, 63].

From the set of collected coordinates, we used Google's API to retrieve the corresponding images from their database. The API automatically returns the best image aimed at the desired coordinate. We excluded any image captured from a distance greater than 25 meters in relation to the desired coordinate to avoid the presence of wrongly selected building facades. This can happen, for example, if the respective street was not visited by Google's camera, but another street close to the desired one was. In these situations, the API would return a photo from another street, aiming in the direction of the desired building further away. We also manually excluded images that would not clearly show the building facade, such as images with trucks or buses covering the front of the building or photos pointing to wrong directions or undesirable places due to displacements by the GPS.

After this process, a total of 2019 images were available for evaluation. As a ground truth is required to validate the computational models' predictions, we requested 10 field agents to label the collected samples according to one of the five scores representing PCI. These experts were also involved in the fieldwork described in Section 3.3.2. Each agent received images at random, and labelled a distinct quantity according to their availability, ranging from over 800 to roughly 100 annotations by each. As a result, the samples received from 0 to 6 labels from different agents, amounting to 1340 unique samples with at least one label. From the provided

candidate labels we derive a definitive label through majority voting. In this process we had to discard 143 samples where the labels were completely disagreeing, with each agent assigning a distinct category, leaving a final set of 1197 samples.

### 3.4 PCINet

Our goal is to leverage the power of deep neural networks to recognize visual patterns from images of facades such that they can accurately approximate the PCI. In other words, we train a deep learning-based model for classification, receiving a facade image as input and outputting a vector of probabilities for all possible PCIs. We adopt a common strategy from deep learning: fine-tuning a pre-trained convolutional network model. The main idea is to transfer knowledge previously learned from a large-scale database, which allows specializing a pre-trained model on a target domain with much less data and training time required [64].

To choose an architecture from the available set of pre-trained models in the literature, we consider the PyTorch framework [65]. It provides an extensive library of model weights trained on ImageNet1k, a large-scale database commonly used as source training, with 1000 object classes from a large variety of categories (e.g., animals, vehicles, and appliances), with some instances of facades for classes such as bakery or boathouse. Fig 2 is a comparison of accuracy (on ImageNet1k) versus the number of parameters for all available models. The number of parameters has a direct impact in computational performance, with more parameters requiring more infrastructure to run. We chose the smallest version of EfficientNetV2, marked in red in the figure, which offers a good trade-off between both measures, achieving over 84% accuracy with a little more than 20 million trainable parameters. Our final model, leveraging EfficientNetV2's architecture pre-trained on ImageNet1k and fine-tuned on our collected dataset, is hereby named PCINet.

Because our database is highly unbalanced between classes, another concern is mitigating the bias it can impose during training, skewing the inferences towards more common classes. We adopt two strategies. First, our model is optimized based on the Focal Loss [66], an optimization metric designed to handle class imbalance and information asymmetry, meaning it can focus on harder inferences, whether the difficulty arises from the lower number of samples from a given class, or on how they differ from the remaining data distribution. Second, we adopt a resampling strategy such that we slightly undersample the most common classes for each epoch. It is worth highlighting that since the training of neural networks takes place over several epochs, i.e., optimization over the entire training set, in each epoch, we randomly load a distinct undersampled subset. The code for PCINet is publicly available at the following repository: https://github.com/patreo-lab/PCINet.

## 4 Results

### 4.1 Descriptive results

After the process described in Section 3.3.3, our database contained a set of 4060 sampled buildings. The relative frequencies of the type of buildings and PCI characteristics obtained are presented in Table 1.

It was verified from the collected data that Campinas predominantly belonged in intermediate and good building conditions, being predominantly in category 4 (37.7%), followed by categories 3 (28.5%) and 5 (19.6%). Only 14.2% of the sampled buildings fell within categories 1 and 2, representing more precariously constructed conditions. We also found that the majority of buildings had completely paved backyards (50.9%). As for the facade conditions, we observed a distribution similar to that collected from the building conditions: a greater frequency in the categories good and intermediate (3, 4, and 5), corresponding to 79.8% of

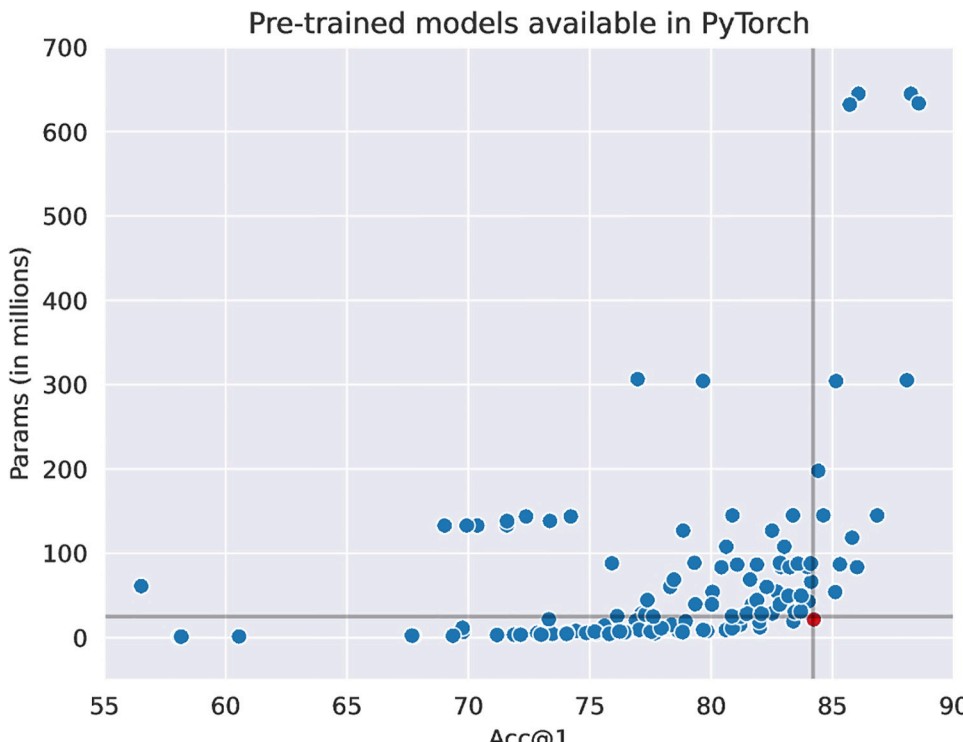

**Fig 2. Scatterplot with the trade-off between accuracy and number of parameters for all pre-trained models for image classification available in PyTorch.** Marked in red is the smallest version of EfficientNetV2.

**Table 1. Distribution of categories of PCI characteristics collected in the field.**

| Target | Distribution of categories | | | | |
|---|---|---|---|---|---|
| Building Type | **House** | **Commerce** | **Industry** | **Apartment building** | **Others** |
| | 84.00% | 10.00% | 0.40% | 1.40% | 4.20% |
| Facade condition | **1** | **2** | **3** | **4** | **5** |
| | 3.60% | 16.60% | 31.70% | 34.60% | 13.50% |
| Building condition | **1** | **2** | **3** | **4** | **5** |
| | 2.30% | 11.90% | 28.50% | 37.70% | 19.60% |
| Backyard condition | **1** | **2** | **3** | **4** | **5** |
| | 3.30% | 11.20% | 28.70% | 37.60% | 19.20% |
| Backyard paving | **1** | **2** | **3** | **4** | **5** |
| | 3.30% | 6.50% | 12.70% | 26.60% | 50.90% |
| Shading | **1** | **2** | **3** | **4** | **5** |
| | 3.10% | 13.50% | 49.80% | 24.30% | 9.30% |
| Roofing | **1** | | **2** | | **3** |
| | 2.30% | | 23.40% | | 74.30% |
| Recipients | **Present** | | | **Absent** | |
| | 48.80% | | | 51.20% | |

**Table 2. Correlation of each PCI characteristic with facade condition.**

| Target | Spearman correlation |
| --- | --- |
| Building condition | 0.8475269, $p-value < 0.0001$ |
| Backyard Condition | 0.7168006, $p-value < 0.0001$ |
| Backyard paving | 0.3813468, $p-value < 0.0001$ |
| Shading | 0.3177191, $p-value < 0.0001$ |
| Roofing | 0.3060127, $p-value < 0.0001$ |
| Recipient | 0.1748162, $p-value < 0.0001$ |

buildings compared to 20.2% in categories 1 and 2, which indicated a worse conservation situation. Most buildings had a partial shading of one-third of the backyard (49.8%) and clay roofs (74.3%). Containers that can be used as breeding sites for mosquitoes were observed in approximately half of the buildings visited (51.2%). Almost all buildings surveyed were houses (84,0%) or for commerce (10.0%). Taking the facade condition as a parameter, we verified how it statistically relates to the other variables measured. We observed that it strongly correlated with building and backyard conditions and had a good correlation with backyard paving and shading (Table 2).

## 4.2 Deep learning-related results

To assess the quality of PCINet, we performed a robust protocol entitled K-fold cross-validation. It consists in splitting the available data into *k* equal-sized random sets, using $k-1$ sets for training, and leaving one out for testing, thus training and evaluating *k* different models. This protocol is more reliable as it allows the assessment of the expected variance in model behavior and avoids skewed metrics due to specificities that might exist in a single random selection of test data. We work with $k = 5$ folds in the following experiments.

The following are the training details necessary for reproducibility. We replace EfficientNetV2's classification head, originally designed for 1000 classes, with a linear layer containing 5 neurons, followed by a softmax activation to produce a vector of probabilities for all five possible facade conditions. Optimization is performed using the ADAM algorithm [67] with a fixed weight decay set to $5e^{-5}$ along with a learning rate scheduling strategy. It consisted of an initialized learning rate of $1e^{-5}$, decreasing this value by a multiplying factor of 0.5 every 10 epochs. We trained each model for a total of 50 epochs. These hyperparameters were empirically tuned to ensure a smooth convergence and avoid overfitting on the training set. Finally, to set the weight for each class required to guide the focal loss, we approximated values inversely proportional to the number of available samples for each facade PCI value, precisely as follows: {4.5, 1.0, 0.5, 0.5, 1.2}. This represents that, for instance, there were nearly ten times more facades where the facade condition was set to 4 relative to 1.

Data collection introduces several challenges into the database such as dirty sensors producing a blurred effect, or camera orientations capturing upside down or otherwise rotated images. Thus, to increase robustness, we randomly apply the following data augmentation techniques while training: horizontal and vertical flips, rotations from the set {90°, 0°, −90°}, and a gaussian blur with kernel size 5 and sigma varying within the interval [0.1, 5]. Along with these transformations, in order to comply with the input settings of the pretrained model, we resize the images to 384 × 384 and apply a z-score normalization with mean pixel $\mu$ and standard deviation $\sigma$ from ImageNet1k.

We can derive the prediction from our model's vector of probabilities output by selecting the facade condition with the highest probability. Based on this, Fig 3 shows the confusion

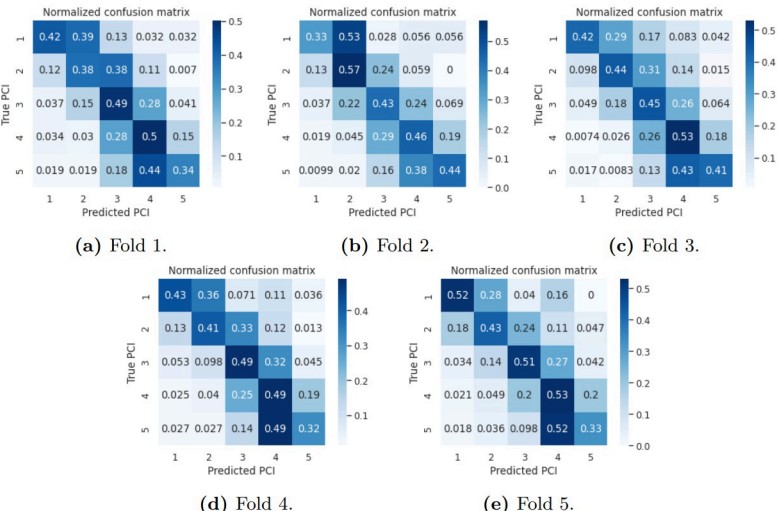

**Fig 3. Classification results from PCINet for all evaluated folds.**

matrices for all evaluated folds. Most noticeably, the matrices always show a thick diagonal, meaning nearby classes are often confused amongst themselves. This is consistent with the fact that classes are strongly related to a 5-point scale of housing conditions from worst to best. This reflects that neighboring classes have similar characteristics, enough to confuse the model, which also indicates a similar issue observed by human agents when assigning labels to the data. However, these errors are not sufficient to compromise the risk assessment of large areas, as mistakenly predicting a building condition by a distance of 1 on the scale has a low impact on risk assessment. These conclusions will be further discussed when looking at the problem from a regression perspective.

Considering the labels as a 5-point scale of intensities, we can derive three metrics to understand our model's behavior. First, the recall and precision for each of the 5 classes allows us to evaluate, respectively, how many of the expected samples of those classes are correctly classified by PCINet and how likely the model is to be right when classifying a sample as the reference class. In this regard, despite our efforts to handle class imbalance, classes 4 and 1 are the best and worst performing ones, respectively; not coincidentally, they are the most and least common of classes. Additionally, we can derive the Mean Absolute Error (MAE) as a distance from the predicted class to the true facade conditions, to look at the results from a regression perspective. The latter aids us in understanding how wrong a given prediction may be on average. Table 3 presents these metrics divided per class as "PCINet Hard Metrics". Although the average recall and precision per class may seem low, usually below 50%, with a high standard deviation (up to 6%), the mean absolute error of predictions is also low, meaning wrongly predicted facade conditions lie within an acceptable error margin. This is related to the fact that the nature of the data and its labels is highly subjective: although it is noticeable the differences between the facade of a low condition building when compared to a high condition one, the same can not necessarily be said when comparing buildings with neighboring levels of condition. This can be viewed in Figs 4 and 5: taking as an example the bottom building correctly classified with PCI label = 2 from Fig 4, although the human annotator classified this building as PCI level 2, it could be argued that this building can be considered of level 1 as it presents characteristics that are common to buildings in this situation, such as lack of plastering and

**Table 3. PCINet metrics divided per class.** Reports are averages over all folds accompanied by the standard deviation in parentheses. Hard metrics represent a strict classification perspective. Soft metrics consider predicted PCI with a distance to the ground truth lower or equal to 1 to be acceptable, as shown in the column valid predictions, where the actual ground truth is highlighted in bold.

| Reference PCI | MAE | PCINet Hard Metrics | |
| --- | --- | --- | --- |
| | | Recall | Precision |
| 1 | 0.937 (+/- 0.073) | 0.423 (+/- 0.059) | 0.248 (+/- 0.017) |
| 2 | 0.693 (+/- 0.104) | 0.445 (+/- 0.066) | 0.482 (+/- 0.044) |
| 3 | 0.619 (+/- 0.041) | 0.474 (+/- 0.027) | 0.484 (+/- 0.045) |
| 4 | 0.579 (+/- 0.039) | 0.501 (+/- 0.027) | 0.507 (+/- 0.036) |
| 5 | 0.871 (+/- 0.067) | 0.367 (+/- 0.046) | 0.373 (+/- 0.033) |
| Reference PCI | Valid Predictions | PCINet Soft Metrics | |
| | | Recall | Precision |
| 1 | **1**,2 | 0.792 (+/- 0.049) | 0.612 (+/- 0.039) |
| 2 | 1,**2**,3 | 0.877 (+/- 0.034) | 0.893 (+/- 0.030) |
| 3 | 2,**3**,4 | 0.905 (+/- 0.014) | 0.930 (+/- 0.010) |
| 4 | 3,**4**,5 | 0.940 (+/- 0.013) | 0.939 (+/- 0.012) |
| 5 | 4,**5** | 0.819 (+/- 0.022) | 0.847 (+/- 0.011) |

painting on the wall. Alternatively, a building facade can also be labeled with a higher level of PCI between two different annotators. This subjective aspect hinders the results of a classifier model, as it makes it difficult to objectively identify the patterns that represent each class, causing the predictions to be commonly mistaken between nearby classes.

Considering the subjectivity in data annotation, analyzing the results with hard metrics from a classification perspective may be misleading regarding model performance and its real world use. Prediction errors between nearby classes, with absolute error (AE) equal to 1, should not be treated equally as prediction errors between further classes ($AE > 1$), as errors committed between nearby classes will hardly impact the identification of higher risk areas in a large-scale scenario. Thus, from a regression perspective, the mean absolute error of PCINet's predictions is $0.66(+/-0.01)$ across all folds and all classes (the values for each individual class can be seen in Table 3). This shows how the prediction errors lie within an acceptable margin that is below 1. In other words, wrong predictions are usually between classes of one level lower or higher than the ground truth.

We further analyze the problem by "softening" labels and predictions. Specifically, we consider predicted PCI with a distance to the ground truth lower or equal to 1 to be acceptable. In this scenario, the accuracy of the model goes up to $89.73\%(+/-0.48)$, decomposed as $46.13\%$ $(+/-1.11)$ of predictions exactly equal to the ground truth (hard success) and $43.60\%(+/-1.04)$ of predictions with absolute error equal to 1 (soft success in which the input was classified as a nearby class). Furthermore, only $1.97\%(+/-0.52)$ of predictions were incorrectly classified with absolute distance to the ground truth greater than 2, which means that "harsh" errors from the model are actually rare. Table 3 presents recall and precision for each individual class in this scenario under "PCINet Soft Metrics". The high values of recall observed in Table 3 are specially important, considering that it is desirable that high-risk buildings are not missed when employing the model. The great improvements in precision under the soft classification scenario once again reinforce how the errors lie between nearby classes. On this subject, the lowest value of precision happening for PCI level 1 indicates that the model is specially subject to false positives for this class. This is a direct consequence of the low amount

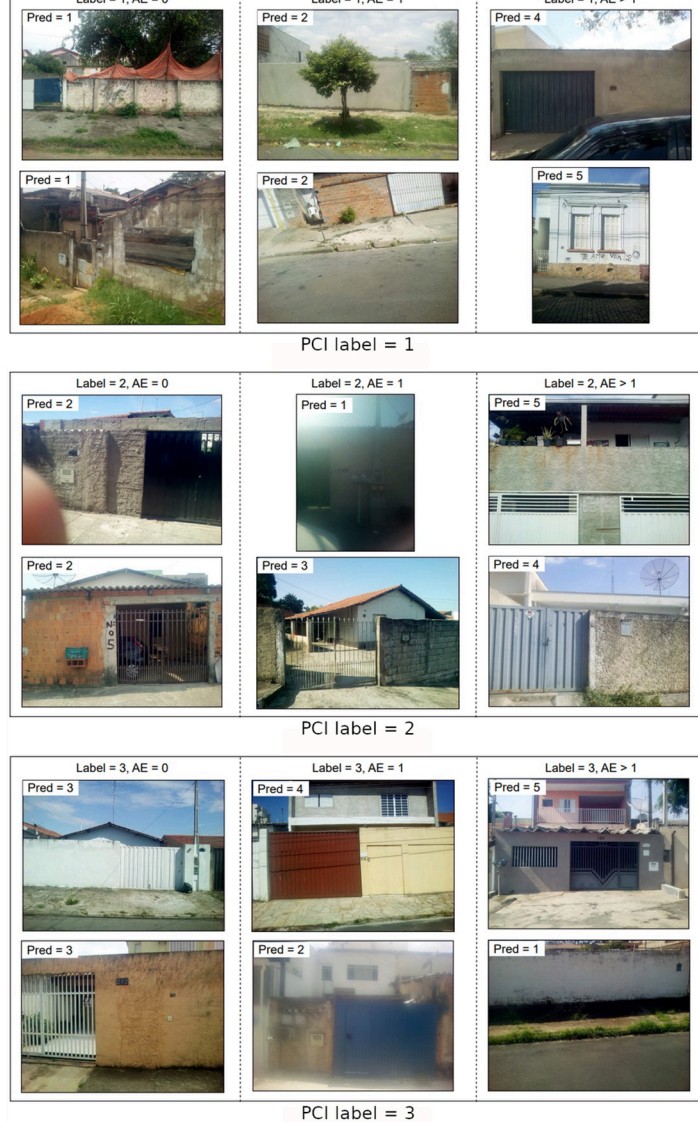

**Fig 4. Random facade samples with different labels for facade conditions.** Images are divided into three columns according to PCINet's ability to classify them, from left to right: correctly classified samples, wrongly classified samples with absolute error (AE) equal to 1, and wrongly classified samples with absolute error (AE) greater than 1.

of samples for this category, which could be improved or avoided with more training data in future applications of PCINet.

To support further discussion on the model's behavior and data-related improvement opportunities, Figs 4 and 5 present a few samples from our dataset subdivided into three columns. Columns refer to three model behavior types: correctly classified samples, wrongly classified samples with absolute error equal to 1, and wrongly classified samples with absolute error greater than 1.

It is worth mentioning that some samples depicted in Figs 4 and 5 were rotated such that all buildings were correctly oriented for better visualization. Although we randomly sampled a small number of images from each type, such visualization surfaces a few essential aspects. For

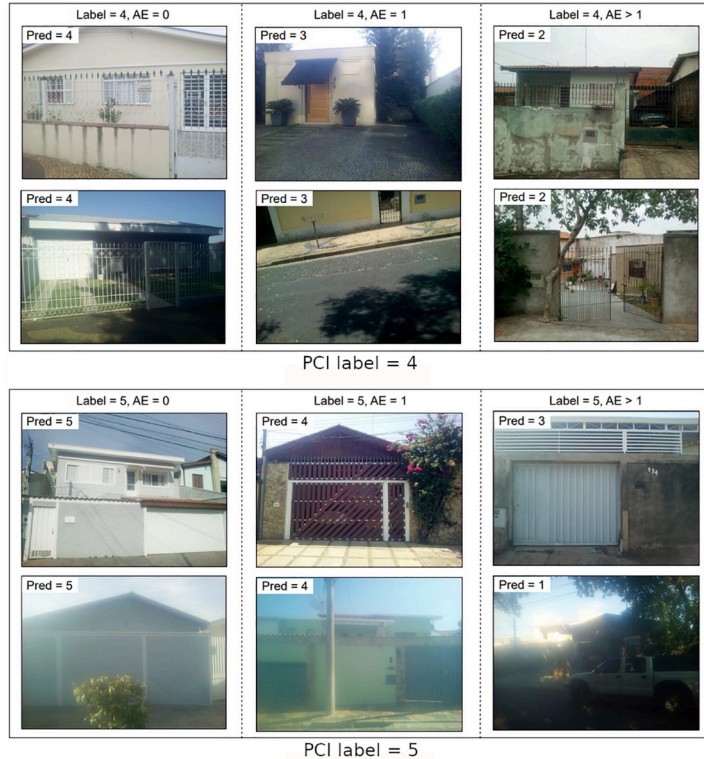

**Fig 5. Random facade samples with different labels for facade conditions.** Images are divided into three columns according to PCINet's ability to classify them, from left to right: correctly classified samples, wrongly classified samples with absolute error (AE) equal to 1, and wrongly classified samples with absolute error (AE) greater than 1.

instance, we can see varying image qualities, with samples presenting distortions such as blur and extreme lighting conditions. These and other characteristics constitute significant challenges for machine learning approaches, which we attempt to tackle by employing data augmentation techniques to randomly distort images while training.

Regarding PCINet's ability to classify facades, we should not draw general conclusions from such a small number of samples depicted in Figs 4 and 5. But it is worth noting that, although images were randomly sampled, we see instances such as the top image with *Label* = 4 and *AE* > 1 in Fig 5 labelled as PCI level 4 but depicting lower level conditions such as the scraped paint and broken pieces of the front wall. We leave as open questions if labels can be objectively and consistently inferred by human agents or whether they are influenced by other aspects such as the neighborhood and remaining characteristics of the building. Along with the aforementioned metrics, we can also discuss whether a 5-point scale is too fine-grained given that PCINet struggles to distinguish neighboring classes. This subjectivity of the problem represents a difficulty that human agents also face in their work.

**4.2.1 Street view results.** Once we understand PCINet's behavior, the following experiment emulates how the model would be deployed in a real scenario, producing a geographic risk assessment over entire neighborhoods. For this experiment, we leverage the data collected from Google Street View. This already constitutes a challenge for our model because images differ in how they were collected, from photographs taken by humans focusing on aspects of interest to an automatic collection from software. Additionally, these facades were extracted

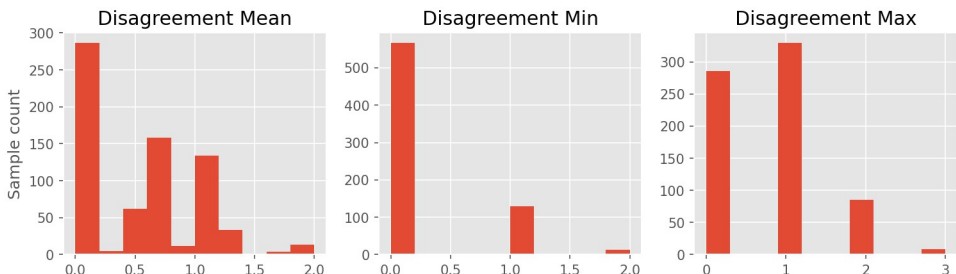

**Fig 6. Histograms of disagreement between labels provided for the Google Street View dataset we collected.**

from neighborhoods never seen in PCINet's training, strengthening our analysis of its ability to generalize to new data.

Before we dive into the results, we measured an agreement rate of the provided labels to evaluate our claim of problem subjectivity. From the complete set of 1340 samples, we selected the ones that were labelled by at least two different agents, amounting to 709 samples. To be consistent with how we measure our own model, the chosen metric was the absolute distance between each pair of labels referring to the same sample. In other words, it is a combination $\binom{n}{2}$ where $n$ is the number of labels for a single sample. For each pair of labels $c_i$, $c_j$, we measure $|c_i - c_j|$. This produced one or more distance measures for each sample.

Fig 6 presents histograms for three metrics derived from these distances: mean, maximum and minimum disagreement rates for each sample. Minimum disagreement shows the 143 samples we dropped during the majority voting process due to their complete disagreement, where 130 had a disagreement of distance 1 and the remaining 13 with distance 2. Maximum disagreement shows that there is only 286 samples with complete agreement amongst field agents. Adding to the 329 samples where at least two field agents disagree by a distance of 1, we have 615 samples (out of 709) where disagreement is below or equal to 1. This is evidence that absolute errors $AE <= 1$ are within an acceptable margin if we consider humans to be the gold standard. Finally, disagreement mean is a distribution with mean and standard deviation values of 0.512 ± 0.494, one more evidence to support our choice of softening evaluation metrics.

Moving on to our PCINet proposition, we leveraged all $k = 5$ models trained on the database collected by field agentes, testing on Google Street View images. To produce a single prediction for each data point we conduct a majority voting process, i.e., extracting the mode from candidate inferences produced by all models. We conducted this process on the final set of 1197 samples described in Section 3.3.5. Fig 7 presents a comparison between human-provided labels for facade conditions with PCINet's prediction. Regions are divided into three types associated with general levels of risk (low, intermediate, and high). This subdivision allows us to visualize PCINet's ability to grasp the general risk tendency for a given region. For instance, Fig 7D shows area 4 (with lower socioeconomic level), where both the labels and predictions overwhelmingly assign low PCI levels to facades. Fig 7C shows area 3 (with an intermediate socioeconomic level), which presents two aspects of interest: (1) while human labels assign medium PCI to a substantial amount of facades, our model tends to assign higher indices to the same regions, and (2) PCINet accurately located clusters of high-risk samples, highlighted in red in the lower left part of the images.

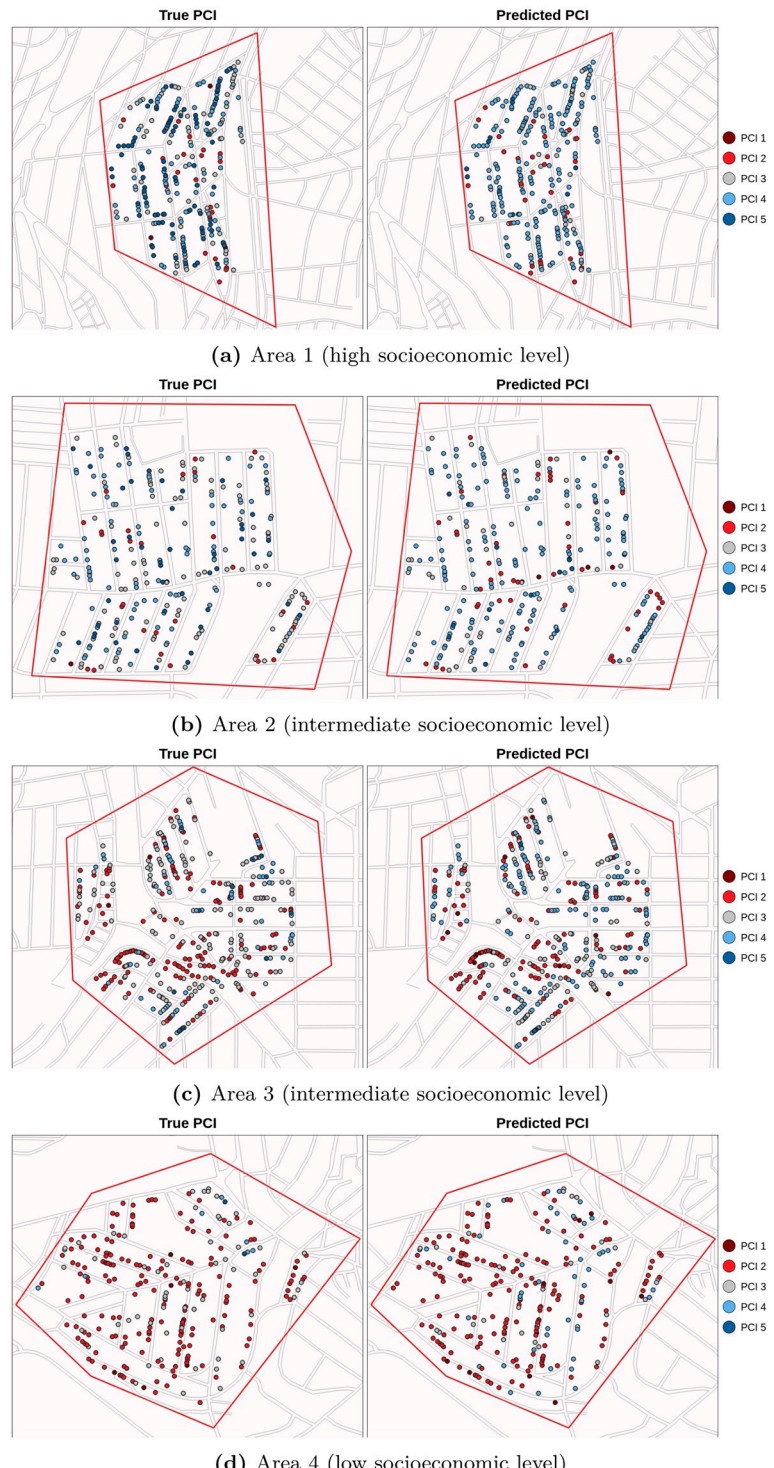

**Fig 7. Geographic distribution of labels and PCINet's predictions on the database of facades collected from Google Street View (best viewed in color).** Each area corresponds to the respective region indicated in Fig 1. Base layer of the figures: https://www.ibge.gov.br/geociencias/downloads-geociencias.html?caminho=recortes_para_fins_estatisticos/malha_de_setores_censitarios/censo_2010/base_de_faces_de_logradouros_versao_2021.

Finally, similarly to how it was done with data captured by field agents, we also evaluated PCINet with the Google Street View dataset in terms of a regression problem and in terms of a classification with "soft" metrics. The mean absolute error was $MAE = 0.57$, similar to the mean disagreement of labels (see Fig 6), once again showing that PCINet errors fall within an acceptable margin. In this scenario, if we consider classification with hard metrics, the overall accuracy is 49.28%. However, by allowing PCI predictions to fall within a distance margin $AE <= 1$, as they do not heavily impact large-scale scenarios and also happen in human annotation, the accuracy goes up to 93.23% (where 49.28% of predictions are equal to the ground truth, plus 43.94% of predictions with $AE = 1$ from the 1197 available samples). Moreover, only 0.41% of the predictions can be considered harsh errors with $AE > 2$, demonstrating that PCINet rarely commits serious mistakes that could impact the process of identifying high risk areas. These results indicate that PCINet is a robust model, capable of being deployed to work with data acquired from different sources.

Despite its limitations, we argue that PCINet is a scalable strategy to triage large areas. The entire process can be automated through data collection from Google Street View and facade condition inferences with PCINet. Its ability to locate high-risk clusters can expedite prioritizing areas for further human inspection. Notably, human agents rely upon a broader set of attributes, such as social and environmental characteristics of different regions; hence, they are better equipped to decide on practical interventions for public health. PCINet is merely a tool to aid in the decision-making.

## 5 Discussion

### 5.1 Assumptions

The strong positive correlation and dependency relationship between the facade conditions and building and backyard conditions, not found in other studies on the topic, along with the positive correlation with backyard paving and shading, shows that the higher the facade condition level, the higher the PCI value, considering its original definition with three categories [26] and its extended version proposed by Barbosa *et al.* [31]. During data collection for the present work, field agents also reported that the resident's care for their property is reflected in the condition of the entire property. That is, if the facade is well maintained, the building is generally well maintained, as it is the yard. Likewise, the opposite is observed; if the facade is poorly maintained, it reflects a worsening of the property's general condition. This is an indicative of the correlation between the facade's condition and the building's general condition. With this, it is possible to say that from the facade PCI, we could infer the general PCI of a building.

The great advantage of this finding is that we could infer the PCI relatively accurately from a single variable, which is also the easiest to collect in field routines, as it is available regardless of some adverse conditions, such as the owner not being at home or not allowing health agents to inspect the building. The facade variable is also one that can often be verified without the need for an agent to go to the field or be verified quickly, allowing the collection of a more significant amount of data in a shorter period, enabling a faster and more economical assessment of the risk of infestation.

Another important premise we assume is that deep neural networks can learn visual patterns related to facade conditions. Our assumption also involved a more fine-grained set of categories, labeling facades with a 5-point scale of indices. According to the results, while neural networks can separate low-condition from high-condition facades with sufficient accuracy for the purposes of risk mapping of neighborhoods, they do not perform well with such granularity of indices. The reported confusion matrices showed thick diagonals, indicating high

confusion rates among neighboring classes, and a measure of mean absolute error of inferences confirms that errors made by our model are within a margin of $+/-1$.

This aspect is worth a discussion regarding the source of such errors. While it can indicate a limitation of our approach, it may also hint at biases during data collection and labeling. Other factors, such as the overall characteristics of a neighborhood or building condition cues other than the facade itself, may influence human agents in the field. This aspect can be assessed in future works by labeling the entire training set with a strategy similar to that used for our Street View test set. If human agents have nothing but the image of a facade to rely on, it may reduce these biases not included in the inputs fed to the neural network. Additionally, the granularity of indices adopted in our work may increase the subjective nature of the assessment. Although human agents receive indications of what constitutes a low building condition (litter, cracks, etc.), there is no objective set of calculations to obtain the final label.

As we have shown our methodology could identify buildings with a higher risk of *Ae. aegypti* infestation, it could be used to optimize the arbovirus disease control program. Therefore, crucial issues are to improve our method and formulate protocols for municipalities interested in applying it. First, we will have to answer whether or not it is necessary to survey a sample of buildings conducting field visits by field control agents to classify their facades. This step could be substituted for digital building facades obtained from Google Street View, among other possibilities for some municipalities. Second, in both cases, it will be necessary to define the sample sizes for different types of municipalities. Moreover, it will be necessary to consider the diversity of building types inside the city. Depending on the characteristics of each municipality neighborhood, different areas, with varying degrees of variability, would require different sampling efforts. We are certain that each situation will require a specific approach and that the results obtained for a given situation cannot be automatically used in a different situation. In trying to translate our results from one to another reality, our models will require adjustment. Nonetheless, as a machine learning approach, our algorithmic core will benefit from trying to represent new situations, allowing its improvement, even though small surveys of buildings will be necessary to visit at the field to validate the modeling in new situations.

## 5.2 Computational modeling

Although neural networks are successful in image classification, their use to predict PCI by exploiting facade conditions (extracted from ground images) is entirely new. Remarkably, given all the specificities of the problem, its modeling, i.e., defining the input data, output, etc., is as important as designing the network architecture as it directly impacts the performance.

In general, our modeling and proposed method showed promising results, capable of identifying risk areas using only ground images without needing to visit all the city's buildings. The conditions for this are related to the positive correlation we found between the facade conditions and the traditional and extended PCI components [26, 31] as well as to the results of previous studies [28–31], showing a good relationship between *Ae. aegypti* infestation and PCI. Supposing we infer the building facade condition level using images from Google Street View or other sources, as our results showed, we would have a reasonable approximation of the building infestation risk level. With this result, we can classify the buildings in risk degrees and select the ones with the highest degree to prioritize and develop vector control activities. The Brazilian arbovirus disease control program establishes a minimum of six visits to all urban buildings of a city during a year [68]. This is unfeasible in mid-sized cities and impossible in large ones [22]. As studies have shown [69–71] that only a minute proportion of the buildings of a city have conditions to support mosquitoes, prioritizing the ones with the highest

probability of being infested by *Ae. aegypti* will allow health services to apply their resources better and achieve better results than those obtained with the current control strategy [68].

One issue to be discussed is better ways to aggregate the buildings by facade conditions to achieve better vector control results. Small cities in Brazil, and probably worldwide, are mixed, with buildings in different conditions occupying the same areas. Buildings in mid-sized and large cities in Brazil, and probably worldwide, are clustered in term of socioeconomic level, type of construction and utility (houses, apartments, commercial and industrial buildings, etc.), and cultural aspects, among other factors. Vector control in small cities can be organized by buildings and developed in the ones with the highest infestation risk level. Meanwhile, in mid-sized and large cities, the control could be organized by block, census tracts, or neighborhood, using the facade average values or proportion of the facade highest values of these areas to prioritize the one to be considered at the highest infestation risk.

### 5.3 Feasibility of deployment

Prevention and control programs for *Ae. aegypti* incur high costs, partly due to their reliance on control methods primarily based on building visits aimed at vector elimination, often requiring extensive operational coverage. These routine control methods involve reducing breeding sites and the use of larvicides and adulticides, resulting in temporary and limited impact on arbovirus disease prevention, especially when coverage is constrained and rarely extends to the entire municipality. Furthermore, these programs are vertical in nature and often do not account for the heterogeneity and diversity of *Ae. aegypti* ecology, including local transmission cycles [72].

A study conducted in a mid-sized Brazilian city revealed that the required coverage for routine control program activities should occur every two weeks [22], a significant departure from the currently recommended schedule in Brazil, which is every two months [68]. Implementing such a frequent schedule would result in an impractical operational cost. In Brazil, a study estimated an investment of 1.5 billion in vector control in 2016, along with an estimated medical cost of 374 million and indirect costs of 431 million, totaling 2.3 billion [23].

Regarding entomological surveillance for *Ae. aegypti*, the primary method relies on larval inspections in domestic breeding sites, such as the Breteau Index and House Index [73], and managers escalate control measures based on these indicators. However, these indices face significant criticism due to their costly nature and dependence on the motivation of field agents to effectively seek out larvae and breeding sites, including those in hard-to-reach areas [74]. Another crucial point is that these indices do not take into account the productivity of breeding sites, and they do not serve as a reliable indicator of adult mosquito density, given that it is the adult female mosquitoes that transmit the disease [14, 75, 76]. In a study conducted in Brazil, no significant variation in the intensity of vector infestation was observed in the evaluated areas. Therefore, it was not a determining factor in the incidence of dengue in the studied municipality [77]. Based on a systematic review, studies have demonstrated the impact of larval population interventions. However, these dengue control interventions, which reduce vector populations, have not shown a clear correlation between this reduction and the risk of disease transmission [78].

Several studies have recognized the high cost of dengue and other arbovirus control programs and their low effectiveness worldwide [23, 79, 80], as we have pointed out. The methodology we developed depends on the availability of digital facade images. The main issue is obtaining digital facade images for socioeconomically deprived regions with higher *Ae. aegypti* infestation risk [34]. The acquisition of images in these areas and others not covered could be done using cars with 3D cameras programmed to collect facade images throughout the city.

The image acquisition from sites or vehicles will represent new costs for the municipalities. These costs would be much smaller than visiting all buildings as this new approach will identify the highest-risk buildings to be visited.

Given the high costs associated with *Ae. aegypti* control and the limited resources in endemic countries, actions should be strategically directed to maximize both effectiveness and efficiency [14, 24]. Consequently, focusing actions on priority areas will lower costs for the *Ae. aegypti* control program [24]. In this study, the use of an AI model to classify building facades using Street View images proved effective. It could be applied to classify buildings and extrapolate this classification to larger areas, such as blocks or neighborhoods. This approach may be valuable for categorizing areas with a higher presence of vector breeding sites because previous studies using PCI have demonstrated that elevated PCI values are associated with a higher likelihood of *Ae. aegypti* breeding sites [31, 81]. Based on the findings of this study, it is believed that the employed methodology can be implemented into the routine of the vector control program. Regarding the improvement of PCI, considering the feasibility of implementing the model used in this study, one possibility would be to include other variables, such as the type and size of existing breeding sites on the buildings and the presence of animals. This implementation could increase the power of predicting risky buildings, allowing this model to replace larval surveys, which, despite indicating the infestation rate and identifying the main breeding sites, often fail to provide quick or localized measurements of mosquito abundance and have a high cost.

Dengue, Zika, and chikungunya are urban diseases that could benefit from our proposed methodology, as *Ae. aegypti* develops in urban breeding sites inside and around buildings [9, 37, 69]. Yellow fever in South America currently occurs in silvatic areas [82]. However, there is a risk of its occurrence in urban areas because *Ae. aegypti* is a vector of this virus in urban areas [83]. The areas identified as high risk for *Ae. aegypti* infestation could be used to conduct vaccination campaigns to increase its coverage.

## 5.4 Strengths and limitations

One of the limitations of the present study was the classification of the facade of buildings used to train the model to identify their characteristics. This classification passed through the eyes of the field agent, who cannot always classify correctly, that is, differentiate small characteristics that differentiate buildings. Considering the values used from 1 to 5, the most significant difficulty lies in the intermediate classifications, with a building that should be classified as 3 eventually being classified as 2 or 4. This subjectivity, for the human eye, implies slight differences in the real classification of the building. Greater investment is needed in this standardization and search for other characteristics, such as comparison with the values of neighboring buildings, which can complement this information so that the model can gain precision.

Meanwhile, this study presents several notable strengths and advantages. To begin with, its multidisciplinary nature contributes to advancements across multiple fields of science, including epidemiology and entomology. Furthermore, the proposed method, which combines artificial intelligence with terrestrial images to predict PCI, presents several specific benefits, including: (i) quicker monitoring, as all that is needed to produce a prediction for a given building is a facade image, a much faster process than sending a public health specialist to visit the building, and (ii) wide coverage, as all buildings in an entire city could have their PCI predicted easily, without the need for local visits. Finally, our study relies on meticulously gathered and highly representative samples collected through exhaustive and rigorous fieldwork.

### 5.5 Opportunities for improvements

Google Street View is one the richest platforms in terms of the availability of ground-level imagery. Still, it does not cover the whole world and often lacks data for smaller cities. For the cases in which it does contain available data, the API allows for collecting a few thousand images (usually up to 28, 500) per month for free because Google gifts 200 dollars monthly per account. For smaller cities or fewer regions, this can be sufficient to employ the proposed approaches with no additional cost beyond the computational resources necessary to run the models. For larger regions, the cost of collecting the images from the platform should be taken in consideration. As for the regions where Google Street View has not visited, it is possible to look for other alternatives, such as KartaView (https://kartaview.org/landing) and Mapillary (https://www.mapillary.com/), which serve similar purposes with different sources for the available images. However, these other platforms are usually more limited than Google; thus, it is improbable (at this time) that they would contain data for desired regions not covered by Street View. Meanwhile, city governments or the public health system can organize to implement data collection for the streets of their respective cities, taking photos from buildings in a faster and cheaper way compared to having agents working to visit each place to analyze their conditions, for example, using cars with 3D cameras, as we have pointed out. This would remove the dependency on external data sources, allowing them to adapt the data collection criteria according to their necessities.

Our study relied on a proxy identifier for mosquito infestation through building condition indices. While this can be beneficial from a broader perspective, incorporating such indices into other socioeconomic-related public assessments, there are other approaches more directly related to the target of our work. For instance, mosquito infestation is strongly correlated with the presence of breeding sites. A growing trend in the literature is to frame the problem as a detection task, leveraging remote sensing techniques to locate potential water retention areas. This is commonly approached as detecting a predefined set of object classes often associated with mosquito breeding grounds in urban areas, such as tires, pools, and watertanks [46, 84]. Still, it may also be framed as a general water retention detection based on the physical behavior of water in both natural and artificial environments [85].

Our methodology could benefit and improve from using satellite images to evaluate building shading and backyard paving levels and evaluate socioeconomic conditions. If we had a better way to predict shading and backyard conditions, we could increase our accuracy in predicting PCI. Housing maintenance is particularly challenging for low-income homeowners [86]. Low-income homeowners often lack the resources to properly maintain their homes, leading to greater health risks [87]. This lack of maintenance by socioeconomically vulnerable people can lead to a building with favorable conditions for the breeding and reproduction of *Ae. aegypti*. Different studies correlate infestation rates with low-income urban agglomerations and vulnerable socioeconomic conditions [36, 37] and some studies point to a correlation with higher rates of dengue infection [59, 60]. Considering that lower socioeconomic conditions favor the breeding of mosquitoes and arbovirus occurrence, the use of satellite images to evaluate the socioeconomic conditions of a given area in real-time [45], along with PCI prediction, could increase the health service skill to identify higher-risk areas and thus optimize surveillance and control, directing efforts efficiently.

## 6 Conclusions

We found that the facade conditions were highly correlated with the building and backyard conditions and reasonably well correlated with shading and backyard paving. PCINet produced reasonable results in differentiating the facade condition into three levels. Although we

began trying to use five levels, the results we obtained are in accordance with the traditional PCI definition, with only three levels. Despite its limitations, PCINet is a scalable strategy to triage large areas. The entire process can be automated through data collection from facade data sources and PCI inferences through PCINet. Although further studies are required to confirm our results, we can hypothesize that using PCINet to classify the building facade conditions without visiting them physically is possible. The good correlations of facade conditions with the PCI components incentivize us to improve our methods to estimate the PCI without conducting physical inspections. Although we have a long road ahead, our results showed that PCINet could help to optimize *Aedes aegypti* and arbovirus surveillance and control, reducing the number of in-person visits necessary to identify buildings or areas at risk.

## Author Contributions

**Conceptualization:** Gerson Barbosa, Valmir Andrade, José Alberto Quintanilha, Jefersson A. dos Santos, Francisco Chiaravalloti-Neto.

**Data curation:** Camila Laranjeira, Matheus Pereira, Raul Oliveira, Camila Fernandes, Patricia Bermudi, Ester Resende, Eduardo Fernandes, Valmir Andrade.

**Formal analysis:** Camila Laranjeira, Matheus Pereira, Raul Oliveira, Gerson Barbosa, Keiller Nogueira, Jefersson A. dos Santos, Francisco Chiaravalloti-Neto.

**Funding acquisition:** Valmir Andrade, Jefersson A. dos Santos, Francisco Chiaravalloti-Neto.

**Investigation:** Camila Laranjeira, Matheus Pereira, Raul Oliveira, Camila Fernandes, Patricia Bermudi, Jefersson A. dos Santos, Francisco Chiaravalloti-Neto.

**Methodology:** Camila Laranjeira, Matheus Pereira, Ester Resende, Eduardo Fernandes, Keiller Nogueira, Jefersson A. dos Santos, Francisco Chiaravalloti-Neto.

**Project administration:** Gerson Barbosa, Camila Fernandes, Keiller Nogueira, José Alberto Quintanilha, Jefersson A. dos Santos, Francisco Chiaravalloti-Neto.

**Resources:** Raul Oliveira, Gerson Barbosa, Camila Fernandes, Valmir Andrade, Francisco Chiaravalloti-Neto.

**Software:** Camila Laranjeira, Matheus Pereira, Ester Resende, Eduardo Fernandes, Keiller Nogueira, Jefersson A. dos Santos.

**Supervision:** Jefersson A. dos Santos, Francisco Chiaravalloti-Neto.

**Validation:** Camila Laranjeira, Matheus Pereira, Keiller Nogueira, Jefersson A. dos Santos, Francisco Chiaravalloti-Neto.

**Visualization:** Camila Laranjeira, Matheus Pereira, Patricia Bermudi.

**Writing – original draft:** Camila Laranjeira, Matheus Pereira, Raul Oliveira, Gerson Barbosa, Camila Fernandes, Patricia Bermudi, José Alberto Quintanilha, Jefersson A. dos Santos, Francisco Chiaravalloti-Neto.

**Writing – review & editing:** Camila Laranjeira, Matheus Pereira, Raul Oliveira, Gerson Barbosa, Camila Fernandes, Patricia Bermudi, Ester Resende, Eduardo Fernandes, Keiller Nogueira, Valmir Andrade, José Alberto Quintanilha, Jefersson A. dos Santos, Francisco Chiaravalloti-Neto.

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
