## [Decision Letter · Decision Letter 0]

28 Feb 2024

Dear Dr. Chiaravalloti Neto,

Thank you very much for submitting your manuscript "Automatic mapping of high-risk urban areas for Aedes aegypti infestation based on building facade image analysis" for consideration at PLOS Neglected Tropical Diseases. As with all papers reviewed by the journal, your manuscript was reviewed by members of the editorial board and by several independent reviewers. In light of the reviews (below this email), we would like to invite the resubmission of a significantly-revised version that takes into account the reviewers' comments. 

We cannot make any decision about publication until we have seen the revised manuscript and your response to the reviewers' comments. Your revised manuscript is also likely to be sent to reviewers for further evaluation.

Sincerely,

Roberto Barrera, Ph.D.

Academic Editor

Paul Mireji

Section Editor

Reviewer's Responses to Questions

**Key Review Criteria Required for Acceptance?**

**Methods**

-Are the objectives of the study clearly articulated with a clear testable hypothesis stated?

-Is the study design appropriate to address the stated objectives?

-Is the population clearly described and appropriate for the hypothesis being tested?

-Is the sample size sufficient to ensure adequate power to address the hypothesis being tested?

-Were correct statistical analysis used to support conclusions?

-Are there concerns about ethical or regulatory requirements being met?

Reviewer #1: The manuscript is well-written and easy to follow the content. The methodology and findings are articulated clearly, providing valuable insights. However, to further enhance the paper's contribution and improve its utility, and relevance to the field, the following revisions are suggested:

Vector control is the only option for reducing the incidence of dengue (from Aedes aegypti) for now, the development of meaningful surveillance methods is crucial.The PCINet model's low performance (less than 0.5 with 5-fold CV) raises concerns. Low accuracy and errors should not be overlooked, as even minor misclassifications can significantly alter risk assessments. The consideration of data augmentation, class weights, and sample weights in PCINet training is recommended. Data augmentation techniques such as rotation, flipping, scaling, and brightness adjustment can specifically address varying image qualities, including distortions such as blur and extreme lighting conditions of the collected samples described (section 3.2.3).

For a more comprehensive assessment of PCINet's practical utility and relevance, incorporating entomological data or disease case incidences into the evaluation is advised. The use of collected adult mosquito data with traps between two fieldworks (mentioned in section 3.2.2) is recommended.

An extra parenthesis in "Facade condition)." in Table 1. Need references to KartaView and Mapillary.

Reviewer #2: -Are the objectives of the study clearly articulated with a clear testable hypothesis stated?

Yes, they are.

-Is the study design appropriate to address the stated objectives?

Yes, it is.

-Is the population clearly described and appropriate for the hypothesis being tested?

Yes, it is.

-Is the sample size sufficient to ensure adequate power to address the hypothesis being tested?

For this initial phase, yes, it is.

-Were correct statistical analysis used to support conclusions?

Yes, they were.

-Are there concerns about ethical or regulatory requirements being met?

No, there are not.

**Results**

-Does the analysis presented match the analysis plan?

-Are the results clearly and completely presented?

-Are the figures (Tables, Images) of sufficient quality for clarity?

Reviewer #1: (No Response)

Reviewer #2: -Does the analysis presented match the analysis plan?

Yes, it does.

-Are the results clearly and completely presented?

Yes, they are.

-Are the figures (Tables, Images) of sufficient quality for clarity?

Yes, they are.

**Conclusions**

-Are the conclusions supported by the data presented?

-Are the limitations of analysis clearly described?

-Do the authors discuss how these data can be helpful to advance our understanding of the topic under study?

-Is public health relevance addressed?

Reviewer #1: (No Response)

Reviewer #2: -Are the conclusions supported by the data presented?

Yes, they are.

-Are the limitations of analysis clearly described?

Yes, they are.

-Do the authors discuss how these data can be helpful to advance our understanding of the topic under study?

Yes, they do.

-Is public health relevance addressed?

Yes, it is.

**Editorial and Data Presentation Modifications?**

Reviewer #1: (No Response)

Reviewer #2: I would like to suggest the authors make their sampled data and the source code of the presented PCINet model fully available.

**Summary and General Comments**

Reviewer #1: (No Response)

Reviewer #2: This study presented a deep learning model to predict high-risk urban areas for Aedes aegypti infestation based on building facade image analysis. The contribution of this study is that it found a close relationship between the facade conditions and the building and backyard conditions, which further suggested that the use of street-level images and the presented deep learning model PCINet could help to improve Aedes aegypti surveillance and control at an urban environment. Overall, this manuscript is well structured and easy to understand. It dose a good work in its result analysis and discussion parts. 

Some minors:

1. The authors can try to explain why there is a strong relationship between the facade conditions and the building and backyard conditions. 

2. The authors mentioned they spend some efforts to collect mosquitos with adult traps. I wonder if they can analyze the relationsip between the mosquito(Aedes aegypti) numbers and the PCIs labled by human, so that the quality of the labled samples can be verified. 

3. I would like to suggest the authors make their sampled data and the source code of the presented PCINet model fully available.

PLOS authors have the option to publish the peer review history of their article (what does this mean?). If published, this will include your full peer review and any attached files.

Reviewer #1: No

Reviewer #2: No
---

## [Editor Report · Decision Letter 1]

17 May 2024

Dear Dr. Chiaravalloti Neto,

We are pleased to inform you that your manuscript 'Automatic mapping of high-risk urban areas for Aedes aegypti infestation based on building facade image analysis' has been provisionally accepted for publication in PLOS Neglected Tropical Diseases.

Best regards,

Roberto Barrera, Ph.D.

Academic Editor

Paul Mireji

Section Editor

---

## [Editor Report · Acceptance letter]

29 May 2024

Dear Dr. Chiaravalloti Neto,

We are delighted to inform you that your manuscript, "Automatic mapping of high-risk urban areas for Aedes aegypti infestation based on building facade image analysis," has been formally accepted for publication in PLOS Neglected Tropical Diseases.

Best regards,

Shaden Kamhawi

co-Editor-in-Chief

Paul Brindley

co-Editor-in-Chief
